# Basic Income and the Status of Women in an Established Gender-Equal Welfare State: Results from the Finnish Basic Income Experiment

**DOI:** 10.3390/ijerph20031733

**Published:** 2023-01-18

**Authors:** Olli Kangas, Minna Ylikännö

**Affiliations:** 1Research Director, Department of Social Research, University of Turku, 20500 Turku, Finland; 2Ministry of Economic Affairs and Employment, 00023 Helsinki, Finland

**Keywords:** gender equality, basic income, empowerment, wellbeing

## Abstract

Debates on the gendered effects of universal basic income (UBI) tend to bifurcate into two opposing views. On the one hand, UBI is seen as a strong incentive for women to stay at home and be permanently locked into their care responsibilities. On the other hand, UBI is seen as a tool for empowerment, increasing women’s autonomy, fortifying their capacity to act, and guaranteeing their individual income and income security. This paper contributes to these debates by asking if UBI enhances women’s empowerment or not. Using the survey data compiled in the context of the Finnish basic income experiment (2017–2018), we compare survey responses from the UBI treatment group (n = 586) and the control group (n = 1047). Our results based on χ^2^ statistics and regression analyses show that, while UBI did not affect employment, it was positively associated with individual capacities and confidence in various aspects of life. However, these empowering effects were universal and did not differ between women and men. Our results indicate that UBI is not a gender equality-related issue in established gender-equal Nordic welfare states. On the basis of our findings, we also argue that the previous academic discussion on UBI and on results from various experiments is too universalising. It does not pay sufficient attention to the national social policy contexts where experiments have been carried out.

## 1. Introduction

In the academic literature, debates on the gendered effects of universal basic income (UBI) revolve around two primary issues: emancipation and employment. Regarding emancipation and empowerment, proponents of UBI argue that as a universal and individual benefit paid directly to each individual, UBI fortifies women’s independence in families and society at large. Through their own incomes, women are liberated from patriarchal power structures. Thus, UBI is seen as a policy measure empowering and giving women the freedom to decide on their own lives [1,2,3,4,5,6,7,8,9,10].

The other aspect, which relates to the previous one, is income security and employment. While traditional social insurance-based income transfer schemes are argued to be biased in favour of men, UBI is argued to favour women instead, especially as it fills the income gap between genders and provides women with protection against poverty [8,11,12]. UBI provides women with the option to choose between employment and care work and to combine them more easily. Advocates for UBI argue that women would use the option to increase their labour market participation. However, there are strong arguments that UBI permanently locks women in their caring obligations at home rather than liberating them to work outside the home, struggle for gender equality, and to have more influence on their own life projects [13,14,15]. Thus, some would argue that UBI would be ‘a trap for women’ [16], in fact squeezing women’s degrees of freedom in their life choices rather than being an emancipatory force liberating them to act and increasing their capabilities in the Senian and Nussbaumian sense [17,18,19].

In this article, we evaluate whether UBI is an emancipatory and capability-fortifying device and, if so, whether it has gender-specific effects. To be more precise, we first discuss whether UBI increases labour market participation and then focus on confidence in the possibility of directing one’s own life. Confidence in own possibilities is used as a proxy for the capability to make choices in one’s life and act accordingly; that is, the interest is in the emancipatory force of UBI in general and for women in particular. In the empirical analysis, we use survey data collected at the end of 2018 as part of the Finnish basic income experiment. 

The structure of this article is as follows. First, we review previous studies and theoretical discussions on the gendered effects of UBI. This is followed by the section on data and methodology. The results of our analyses are presented in the penultimate section, followed by a discussion of the findings to conclude the paper. 

### 1.1. Basic Income and Gender in Previous Studies

The impact of UBI on gender or gender equality has received a great deal of attention in UBI debates. While the lion’s share of the discussion has been theoretical, there are some analyses that derive their argumentative power from existing UBI experiments. The common point in these debates, be they empirical or theoretical, is that they seem to bifurcate into two opposing discourses. Regarding the labour market (labour demand and supply) and wider gendered emancipatory effects, there are substantial disagreements between the proponents and opponents of UBI. 

In one stream of the literature, an unconditional income independent of paid work is seen to enhance employment: people can opt for low-paying jobs, start their own small-scale businesses, and be creative when UBI guarantees their basic security. This applies to women, in particular when they can better combine care and (part-time) work. Thus, women’s agency in families, households, workplaces, and communities, which particularly benefit those facing multiple and intersecting forms of discrimination in these spheres of life, would be improved [6,7,8,9,10]. In other words, UBI empowers women, increases their capabilities, and further increases gender equality.

These positive impacts can be more easily generalised in societies with more traditional attitudes and institutionalised practices towards women’s and men’s roles in society, thereby resulting in gendered inequalities in both the private and public spheres of life. Women’s financial independence remains far from being self-evident in many developing economies. Without an established social security system, women are often condemned to live in poverty and at the mercy of men’s benevolence. This brand of discussion is fortified by references to empirical results from India, Kenya, or Namibia [9,20,21,22]. 

However, the possible “women-friendly” effects of UBI are not limited to developing countries but can be observed in developed societies as well, such as European welfare states, in some of which traditional views on gender roles are still quite strong [23,24]. Without accessible childcare and adequate support for mothers to balance childcare duties with work, women are obligated to make choices between the public and private spheres of life [25,26,27]. They have to choose between financial independence and dependence on their spouses, which is often used to justify the introduction of basic income.

The pro-UBI argument is that UBI would make it easier to reconcile work and family life when opening avenues for part-time, sporadic, and short-term employment while simultaneously providing basic financial security. This is consistent with the views of scholars who acknowledge that UBI provides both recognition of previously unpaid care work at home and basic security that provides a basis for all forms of labour market participation [20]. Thus, UBI fills the gap between recognition and redistribution [28]. Of course, this requires part-time or gig-type jobs to be available in the labour market. Thus, the labour market context and institutions matter.

In the alternate strand of the literature, UBI is seen more negatively (for example, [13,14,15]). For this position, it is feared that UBI forms a strong incentive for women to stay at home, locked in their traditional caring positions. Empirical results from the negative income tax experiments conducted in the U.S. and Canada (Mincome) in the 1970s and the 1980s displayed diminishing female labour market participation: wives and single mothers reduced their labour supply, corresponding to three weeks of full-time employment [29,30,31,32]. When given the possibility, women, especially those in lower socioeconomic groups, opted out of unpleasant jobs and chose to spend more time with their children. Thus, the income transfer introduced in the experiments decreased the female labour supply. This finding is, to some extent, corroborated by a Swedish study on the labour market behaviour of lottery winners who reduced their working time as well as those of their spouses [33]. 

More detailed analyses from the U.S. experiment suggested that the reduction in labour supply was strongest among mothers with small children who were a part of the experiment, given that the opportunity to stay at home with their babies was not possible under the existing social security schemes [29,30]. The Canadian Mincome experiment displayed fewer substitutional effects [29], which is explained by the fact that the family leave systems in Canada are better than those in the U.S. Thus, the context of social policy matters.

There have been attempts to bridge the gap between the two opposing lines of argument. While UBI, when generous enough, provides women with the opportunity to choose between family and work, it simultaneously reshapes patriarchal gender norms around paid work and unpaid care, especially as no person is relegated to being a masculinised ‘breadwinner’ or feminised ‘caretaker’ to attain income security. Combined with the provision of affordable quality care services, UBI could potentially facilitate a more equitable distribution of care between genders [5,6,10,34,35,36]. 

In most cases, the motivation to introduce UBI is related to perceived problems and shortcomings in existing welfare systems: holes in the safety net, poverty, overly low take-up rates, and stigma attached to means-tested programs. Furthermore, it is argued that social insurance-based transfer schemes typical for most welfare states disproportionally benefit males with longer working careers and higher incomes compared to women with care duties at home and longer absences from the labour market. While social insurance-based social security schemes provide inferior benefits to those who do not participate in full-time formal employment, UBI would treat everyone equally, regardless of participation in the labour market or length of working career [34,35,36,37]. However, there is an obvious weakness in this argumentation. The further the equality of women and men has progressed in the labour market, the weaker this motivation for basic income. This applies first of all to gender-equal welfare states. Thus, our main hypothesis is that in a well-established welfare state, UBI would not be a strong policy measure for female emancipation. 

In our case, emancipation is rated as empowerment, that is, people’s capacities for action in their own lives in line with Sen’s and Nussbaum’s theories on capabilities [17,18,19]. According to them, people must have the capabilities to master their own lives and participate customarily in the society they live. Nussbaum [19] separates two different forms of capabilities, while the first one, i.e., the internal capability, is linked to the individual’s own agency, and the second one, ‘combined capability’, is related to societal institutions enabling individuals to use their internal capabilities. The proponents of UBI argue that UBI is a policy measure that fortifies both forms of capabilities. Thus, UBI is a device for emancipation. Our main goal is to analyse if that hypothesis holds or not. We ask:(1)Whether UBI enhances emancipation?(2)Whether UBI enhances employment?(3)Are there gender-specific effects?

UBI has not previously tested in a well-developed welfare state with residence-based basic social security, comprehensive social insurance programs and universal social services. The Finnish experiment was the first one in such a context. There are no previous studies that have analysed Finnish experimental data to decompose the possible gender effects. 

### 1.2. Finland—An Established Gender-Equal Welfare State

Nordic countries, of which Finland is the focus of this article, are generally considered model countries for gender equality [38]. According to the United Nations’ and World Economic Forum’s rankings, Finland is one of the most gender-equal countries worldwide [39,40]. 

The cross-cutting principle in gender equality for all policy areas has shaped Nordic societies in a way that women can realise themselves in different areas of life independent of men [24,25,26,27]. Typically, social benefits are paid to individuals rather than to families. Taxation is individual and not family bound. Most importantly, gender equality is promoted through services that enable mothers (and fathers) to reconcile their families and work. Accessible and affordable public childcare is the cornerstone of gender equality, as it presents women with the same possibilities of participating in working life as men. 

In principle, women and men are assumed to be equal in both private and public spheres of life. While families can decide for themselves if the mother stays at home with small children, she is not expected to do so. The opposite is, in fact, the case. Similar to mothers, fathers are increasingly encouraged to participate in childcare and household duties. [24,25]. The internationally extensive family leave system enables mothers and fathers to reconcile family and work, whereas the subjective right to return from parental leave to the same work signals to parents and potential parents that unpaid work at home is valued. 

In Finland, social benefits are typically universal, individual, and non-gendered. For example, unemployment benefits, both basic benefits and income-related benefits, are paid according to the same principle to women and men, and the spouse’s income does not affect the benefit. Some income transfer schemes previously targeting women have historically played an important role in promoting gender equality. An example of this type of ‘woman-friendly’ social policy scheme is the universal child benefit system introduced in Finland in 1948. From the beginning, it was paid to the mother until the child was 17 years old. The amount of monthly child benefit per child is not high, but it nevertheless provides women with financial freedom, and in that sense, is comparable to UBI.

Of course, the promotion of gender equality has opponents, even in Finland. Some of the policy measures taken have weakened rather than improved gender equality and have emphasised the role of women as the holders of primary careers in families. An example of this type of scheme is the child home care allowance, a cash-for-care benefit that is payable to families with children under three years of age in case they choose home care for the children instead of public or private daycare. Hiilamo and Kangas refer to child home care allowance as a trap for women because it has made care work at home more attractive than low-paid strenuous work in particular [16]. The negative impact of this benefit on women’s employment has been documented in many studies [16,41,42], although, without the numerous other policy measures aimed at increasing gender inequality, this impact would be much stronger and would concern a larger share of mothers.

Koslowski and Duvander [43] discussed UBI and gender equality in Sweden from a theoretical perspective. They asked whether UBI has empowerment effects and whether such effects would be different among men and women. Unfortunately, they were unable to empirically test their ideas. However, this is possible in Finland, where basic income was experimented with from 2017–2018. When the Finnish centre-right government decided to start the UBI experiment in 2015, its interest was ultimately in the employment effects of UBI. The secondary interest was in the well-being effects of UBI [44,45]. 

Based on previous research, we already know that in the Finnish experiment, UBI had neither positive nor negative effects on employment when looking at all receivers of UBI [40]. In this study, the interest is specifically in the gendered effects of UBI on employment and in how UBI is potentially linked to recipients’ perceptions of their possibilities and capabilities. 

## 2. Data and Methods 

The Finnish basic income experiment was running for two years, from 2017 to 2018. The treatment group consisted of 2000 randomly selected unemployed jobseekers aged between 25 and 58 years who received unemployment benefits from the Social Insurance Institution of Finland (later referred to as Kela) at the end of 2016 [44].

The Finnish unemployment benefit system is a two-tier system: members of voluntary unemployment funds are eligible for earnings-related unemployment benefits paid by the unemployment fund in question. Those not eligible for earnings-related benefits receive a flat rate unemployment benefit from Kela unless they fail to fulfil the criteria for the benefit, in which case they are eligible for social assistance [46].

The 2000 unemployed who were selected to participate in the experiment (i.e., the treatment group) each received a UBI of EUR 560 net per month for two years. This sum corresponds to the net level of the basic unemployment benefit. The difference between UBI and unemployment benefits was that earned income did not decrease UBI, whereas unemployment benefits were inversely related to income from employment. The rest of the unemployed (about 170,000 persons), who simultaneously received unemployment benefits from Kela, formed the control group. Because of random sampling, the two groups were identical at the beginning of the experiment. 

For the receivers of UBI, the experiment was obligatory to avoid selection bias typical for voluntary experiments. To observe the UBI experiment’s effects, several types of data were collected, including administrative register data, surveys, face-to-face interviews, and media discussions [44]. 

In this study, we utilised survey data collected at the end of 2018 from the participants of the experiment and members of the control group of 5000 individuals. The would-be respondents first received an information letter regarding the survey. A phone-based survey was conducted by the end of 2018. Unfortunately, the response rates remained low, with 31% in the treatment group and 20% in the control group. Owing to the low response rates, we weighted the data to correct for possible no-response bias. The reweighted data were compared with the background characteristics of the original target groups, and no significant differences were found. [44]. 

We analysed six models. The first (Model 1) is used to explain the employment status of the respondents, and the second (Model 2) explains their confidence in finding work. In the survey, the respondents were asked about their labour market status at the end of the experiment. In our data, the value of 1 for the binary variable ‘employed’ corresponded to those respondents who said that they were either employees or self-employed. The other respondents had a value of 0; that is, those who said that they did not have a job and those who said that they did not know if they had a job. Of all the employed respondents, 94% were employees, and the rest were self-employed. 

The respondents were also asked whether they believed they would find a job in the next 12 months if they lost their job or if they remained unemployed. There were three options to answer: yes, no, and cannot say. Overall, 57% of the treatment group respondents and 48% of the control group respondents stated that they believed they would find a job that matched their profession or work experience. Furthermore, 16% of the respondents in the treatment group and 13% of the respondents in the control group chose the option ‘cannot say’. For analytical purposes, the answers were coded dichotomously: 0 = the respondent does not have confidence in finding a job, those who said that they do not know, and 1 = the respondent has confidence in finding a job. 

While the Finnish basic income experiment only included unemployed job seekers, we were unable to evaluate the possible substitution effect, that is, whether distributing UBI decreased the labour supply. We can only assess whether UBI was beneficial for employment, which proponents of the UBI emphasise. In Model 2, we were interested in respondents’ empowerment. Empowerment was also analysed in all models from 3 to 6. 

In the survey, the respondents were asked, ‘How do you feel the following things have developed in your life within the last two years? Confidence in (1) your own future, (2) your own economic situation, and (3) your ability to cope with difficult life situations. The answers to the questions were (1) bad, (2) fairly bad, (3) neither bad nor good, (4) fairly good, (5) good, and (6) I do not know. In the subsequent analyses, the last alternative was coded as a missing value. After modelling the three confidence variables separately, we merged them into one additive ‘empowerment’ index that varied between a low value of 3 and a high value of 15. To test the consistency of the new variable, we ran a factor analysis producing a single factor with high loadings for all three variables (future = 0.887; economic situation = 0.855; coping with difficult life situations = 0.821). Consequently, the Cronbach Alpha was very high (0.821), indicating that the index was internally consistent. 

As independent variables in all models, we first have a binary variable that describes participation in the experiment (0 = member of the control group, 1 = member of the treatment group). In addition, we included gender, age, educational level, and subjective assessments of individual work ability in our models. Work ability was measured through the following question: ‘Let us assume that the top rating for one’s ability to work is 10. How would you rate your ability to work on a scale of 0 to 10, where zero is a very poor ability to work and 10 is an excellent ability to work? The variable is a proxy for respondents’ self-rated health status. 

For analytical purposes, we recoded the variable on the ability to work into five categories instead of 11 for the original variable. In the first category, we combined the first three categories (0–2) into one category. For the next four categories, we combined two categories of the original variable into one (3–4 = 2nd category; 5–6 = 3rd category; 7–8 = 4th category; and 9–10 = 5th category). Age was recoded into six categories (27–35; 36–40; 41–45; 46–50; 51–55; and 56–61 years of age). The variable describing educational level had six categories as well: 1 = basic, 2 = vocational, 3 = high school, 4 = college, 5 = applied university, and 6 = university degree.

We began our analysis with simple cross-tabulations, wherein the evaluation of the significance of differences between the groups was based on the χ^2^ -test. Thereafter, we employed logistic regression in the models with dichotomous dependent variables and univariate general linear models in the models with Likert-scale or continuous dependent variables. 

## 3. Results

### 3.1. Employment and Confidence in Finding New Work

At the end of the experiment, employment was somewhat higher in the treatment group than in the control group, but there were no significant gender differences in the employment rates between the treatment group (35% of women vs. 34% of men reported employment in 2018; χ^2^ sig. = 0.443) and the control group (27% vs. 29%, respectively; χ^2^ Sig. = 0.221). Thus, our survey-based results are fully consistent with the results of the register-based analyses conducted in connection with the evaluation of the experiment [36]. 

One argument in favour of the UBI is that by giving people basic economic security, they are able to combine care work at home with part-time employment, as discussed above. In Finland, part-time work is less common (14% of the total employment) than the EU average (22%). This applies to both women (17% in Finland vs. 36% in the EU-27) and men (14% and 17%, respectively) [47]. Less surprisingly, the gender differences in part-time employment in Finland are clearly smaller than the EU average (4 percent points vs. 27 percent points difference). 

Compared with the Eurostat figures, the share of part-time workers in our data is higher in both the treatment and control groups, indicating that part-time work is a typical route out of unemployment. Furthermore, part-time work is more typical in the treatment group (38% of the employed) than in the control group (31% of the employed), indicating that UBI may have positive effects on part-time employment. In the Finnish experiment, the recipients of UBI, unlike those in the control group, were allowed to maintain the entire benefit, even if they had income from work or entrepreneurship. Hence, the incentive to accept even a very low-paid job was seemingly higher than in the scheme where benefits are progressively taxed away from earnings. 

Regarding gender and part-time employment, 43% of women and 33% of men in the treatment group worked part-time (χ^2^ Sig. = 0.103). The difference was significant in the control group (38% vs. 26%; χ^2^ Sig. = 0.012). This result indicates that UBI (in the Finnish experiment) encouraged men, especially, to receive part-time work instead of full-time employment or unemployment. This reduced the gender gap in part-time work and increased gender equality. Again, the results should be treated with caution because, in the case of women, the underdeveloped part-time labour market in Finland may be an obstacle to discovering a higher increase in part-time work than is observed in the data.

Part-time work is taken either because it is preferred to full-time work or because there is no full-time work available. In our data, 70% of female and 74% of male part-time workers in the treatment group expressed a desire to work full-time instead of part-time (χ^2^ Sig. = 0.453). The shares in the control group were 50% and 75%, respectively (χ^2^ Sig. = 0.015). This indicates that part-time work is undertaken because full-time work is unavailable rather than because full-time work is preferred. This result is not surprising, as household income remains low despite part-time income if the UBI level is low, as in the Finnish experiment.

In the subsequent analysis (due to small numbers in specific groups), we do not look at part-time work separately but at the general level to determine the effects of UBI on employment. In Table 1, the binary outcome of employment at the end of the experiment is regressed on the independent variables, as defined above. As Model 1 shows, UBI somewhat increases the probability of re-employment (the ‘probability’ is about 1.3 times higher among the treatment group compared to the control group), but the coefficient is statistically not significant. This result is consistent with previous register-based studies displaying slightly higher employment rates in the treatment group compare with the control group [45]. In our model (Model 1), gender neither becomes a statistically significant explanatory variable nor the interaction term (Treatment*Gender) between gender and treatment. Unsurprisingly, employment was positively and significantly explained by the level of education and ability to work. Age reduces the likelihood of employment. 

The participants of the Finnish UBI experiment had often been unemployed for extended periods, which indicates that many of them had either health problems or other obstacles in finding new work, such as insufficient or outdated vocational skills [48]. This is also indicated by the highly significant coefficients of educational attainment and workability. The right-hand panel in Table 1 (Model 2) shows whether respondents have confidence in finding work in the following 12 months. In addition to the independent variables in the first model, we included employment status at the end of the experiment to control for the possible effects of an existing job on the dependent variable. 

Receiving basic income increased respondents’ confidence in re-employment (Model 2). The probability of being confident in one’s own re-employment was approximately 1.6 times higher among the treatment group than in the control group respondents. Similar to Model 1, gender does not become a statistically significant explanatory variable, nor does the interaction term between gender and treatment; for both groups, being employed at the end of the experiment significantly increased confidence in finding work. Additionally, better work ability increases confidence in finding work. However, higher age predicts less confidence in finding new work, which is not surprising given that age discrimination is rather common in the Finnish labour market. 

To summarise our results thus far, UBI did not have a significant effect on employment in the Finnish UBI experiment, but it significantly increased confidence in re-employment and, in that sense, empowered the receivers of UBI. Gender and the interaction between gender and treatment do not appear to be statistically significant explanatory factors in either Model 1 or 2. Thus, according to our results, UBI does not have different effects on women and men in an established gender-equal welfare state such as in Finland. 

### 3.2. Self-Confidence as a Proxy for Empowerment

As shown in Table 2, the differences in opinions on all three aspects of confidence between the treatment and control groups are statistically significant. For all the respondents, the highest level of confidence is found in coping with difficult life situations: 67% of the respondents in the treatment group and 58% in the control group answered that they had high, that is, good or very good, confidence in coping with difficult life situations. The lowest level of confidence was found in economic situations. In the treatment group, 44% had high or very high confidence, whereas the corresponding proportion in the control group was lower (33%). The low shares in both groups indicate that the level of the Finnish flat-rate unemployment benefits is too low, and hence, the level of UBI in the Finnish experiment, being set to the level of basic-level unemployment benefits, was too low to satisfy everyday financial needs. However, the higher level of confidence in one’s economic situation in the treatment group indicates that an unconditionally paid social benefit reduces the financial stress caused by low household income [49].

On the one hand, comparing the opinions of women in the treatment and control groups, we find that in all three confidence-based aspects, those in the treatment group reported significantly higher levels of confidence (χ^2^ Sig. = 0.003 for economy; 0.002 for future; and 0.005 for coping) than those in the control group. For men, the corresponding significances were 0.000, 0.042, and 0.207. On the other hand, when we compared men and women in the treatment group, we did not observe significant gender differences (see Table 2). UBI possibly increases confidence and furthers empowerment, although its effects are gender-neutral.

Table 3 presents the results for general linear models wherein the three confidence factors are regressed separately in terms of treatment, gender, the interaction term of treatment and gender, respondent’s age, educational attainment, and the subjective evaluation of one’s ability to work. Because the magnitudes of the estimates are not relevant in this context, in Table 2, we only present the significance of the estimates in each model. Are the interactions (Treatment*Gender) between treatment and gender significant or not? 

For the three aspects of confidence and the merged variable empowerment, the coefficient for treatment was statistically significant, indicating that receiving UBI significantly increased empowerment. In addition, gender becomes statistically significant for explanatory factors in the models (except for confidence in coping with difficult life situations), indicating that there are differences between women and men in coping mentally with unemployment. However, the interaction term between gender and treatment was not significant. Thus, UBI does not seem to have gender-specific effects on empowerment.

## 4. Conclusions

The academic debate on the relationship between gender and UBI is highly divided. The proponents of UBI regard it as a device for increasing social justice, enhancing gender equality, empowering women, and reconciling family and working lives. UBI is regarded as a vehicle for supporting part-time and atypical work, thus expanding employment possibilities in general and for women in particular. Opponents of basic income tend to deny almost all the positive attributes that advocate the presence of UBI. They envisage UBI as a dangerous policy option that, instead of abolishing traditional gender roles, de facto fortifies them and traps women at home by consolidating traditional gender roles.

In their article, Koslowski and Duvander [43] ask, on a theoretical level, without the possibility of empirical verification, whether UBI generates empowerment, and if the answer is affirmative, whether these effects are gendered. Our analyses, which were based on a survey collected in a nationwide Finnish, randomised, and obligatory UBI experiment, sought to answer these questions empirically. 

Regarding employment, we found no significant gender effects, although those respondents who received a basic income had stronger confidence than the respondents in the control group in their ability to find employment. However, this belief did not materialise in higher employment rates. Furthermore, our results indicated that the recipients of UBI were more confident in their futures, abilities to cope with difficult life situations, and possibilities to improve their economic situations. In other words, UBI positively affected recipients’ self-confidence, thereby enhancing their internal and combined capacities [19]. This effect was the same for men and women. 

Thus, we conclude that UBI may be a tool for empowering its recipients, albeit in an established gender-equal welfare state that concerns women and men alike. While UBI is not a universal silver bullet of any kind in enhancing gender equality in a society such as Finland, it may well be that in developing countries or countries with greater gender inequalities. In a highly egalitarian Nordic country, a single change in the benefit system does not affect gender equality. Achieving equality required and still requires decades of development of the benefit-and-service system based on the overarching principle of equality. In sum, the results are sensitive to the context of where the experiment takes place. 

### 4.1. Too Universalising Discussions

On the basis of our findings, we argue that the previous academic discussion revolving around UBI is too universalising and does not pay sufficient attention to national contexts. Theoretical reasoning has been universalised in two ways. First, it has often been based on philosophical argumentation without links to any specific social context or welfare state. Second, if it has some links to existing welfare state models, it most often has been Anglo-American, that is, the liberal welfare state [50], which has formed an institutional context wherein the would-be UBI is submersed (however, see [9,21,22,43]). Nevertheless, the results have been, almost without exception, interpreted to apply universally to all welfare states. 

Each welfare state is a unique institutional entity with its own historical and political background and, consequently, its own institutional context. Thus, we can assume that expected beneficial or detrimental gendered outcomes are different between a country with low gender equality and an established gender-equal welfare state, such as Finland. The institutional context is important, and institutions travel badly. Therefore, the results of the experiments are not directly transferable to other countries.

However, we, too, end up with a few universalising comments or, if you like, lessons from the Finnish UBI experiment. In all societies, minimum income protection, either UBI or Finnish residence-based universal basic security, is a necessary condition for fulfilling the grand goals advocated by the protagonists of the basic income concept. However, this is not a sufficient condition to achieve these goals. In addition, we need an amplitude of family policy measures, such as available and affordable childcare, and we need well-functioning healthcare, education, and employment services to support people and societies in their path towards gender equality. Hence, gender equality cannot be achieved by a single social policy measure. The achievement of more equal societies requires several measures that consistently seek to promote that value. A decent level of money is a necessary but insufficient condition for empowering people. 

As Koslowski and Duvander [39] outline, ‘Money, a floor to stand on, surely helps [to procure freedom], but it will not alone be likely to challenge norms, for example, around parenting practices’. When we want people to change their behaviour to meet shared societal norms, we need different policies that are sufficiently effective to change the practices that produce gender inequality. 

### 4.2. Limitations of the Study and Future Research

The strength of the Finnish UBI experiment was that it was a large-scale, randomised and obligatory field experiment with a treatment and a control group. At the beginning of the experiment, those two groups were identical. There are limitations, too. The experiment lasted only two years which is too short of a period for people to make long-term decisions. There are obvious strengths in randomised field experiments. The flip side of such experimental designs is that we cannot analyse possible saturation/community effects, i.e., what would happen if everybody in the community received basic income [3,20,51,52].

The target group in the Finnish experiment was unemployed. Therefore, we cannot estimate substitution effects, that is, the extent to which men and women would decrease their labour supply if they received UBI. However, the results on the impact of childcare allowance on employment indicate that UBI would possibly produce gender-biased substitution effects. The utilisation of cash-for-care schemes in childcare provides arguments against UBI’s potential to enhance gender equality in the labour market. There is strong evidence that the cash-for-care home care allowance typically used in Finnish families with small children reduces female employment in both the short and long term ([41,42], see also [53]). Whereas on the basis of our data, we can only draw indicative conclusions about the positive impacts that UBI may have on empowerment and capability formation in an ‘established gender-equal welfare state’, we cannot estimate possible negative impacts upon labour supply or any other substitution effects UBI might cause. 

The Finnish UBI experiment produced results that apply to the Finnish context or contexts similar to the Finnish one. It is important to carry out basic income experiments in different surroundings: In developing economies (e.g., the Kenya experiment), where any social benefit introduced will eventually lead to better income, higher social security and well-being, and in more prosperous economies with developed and more comprehensive social security systems (e.g., in the U.S., Germany, Catalonia). Comparison of results from different experiments and experimental designs would yield more reliable knowledge on the possible positive and negative effects of UBI. 

## Figures and Tables

**Table 1 ijerph-20-01733-t001:** Binary logistic regressions on gendered employment impacts and confidence in finding employment in the following 12 months.

Variables	Model 1: Employment			Model 2: Confidence in Finding Employment		
	B	Sig.	Exp(B)	B	Sig.	Exp(B)
Constant	−3.858	0.000	0.021	−0.471	0.213	0.625
Treatment	0.245	0.159	1.277	0.490	0.013	1.632
Gender	0.108	0.469	1.114	−0.314	0.039	0.731
Treatment*Gender	−0.044	0.854	0.957	−0.032	0.905	0.969
Age	−0.013	0.021	0.987	−0.033	0.000	0.967
Education	0.129	0.000	1.138	−0.045	0.228	0.956
Ability to work	0.736	0.000	2.087	0.567	0.000	1.762
Employed	--	--	--	0.961	0.000	2.614
Nagelkerke R sqr.	0.183			0.274		

**Table 2 ijerph-20-01733-t002:** Confidence in economic situations, the future, and coping in difficult life situations among women and men in the treatment and control groups. Share of those respondents with high or very high confidence (%).

Confidence in	Total Sample			Treatment Group			Control Group		
	Treatment Group	Control Group	χ^2^ Sig.	Females	Males	χ^2^ Sig.	Females	Males	χ^2^ Sig.
Economy	44.2	33.0	0.000	47.7	58.3	0.334	33.4	51.5	0.100
Future	61.1	49.3	0.000	67.4	55.4	0.059	54.1	46.1	0.168
Coping	67.3	57.9	0.001	70.8	64.2	0.361	59.6	56.4	0.160

**Table 3 ijerph-20-01733-t003:** General linear model on confidence in coping with difficult life situations, in the future, in the economic situation, and on empowerment (significance of coefficients).

Variable	Model 3: Confidence in Coping with Difficult Situations	Model 4: Confidence in Future	Model 5: Confidence in the Economic Situation	Model 6: Empowerment
Treatment	0.001	0.000	0.000	0.000
Gender	0.147	0.000	0.028	0.004
Treatment*Gender	0.995	0.910	0.408	0.769
Age	0.069	0.002	0.011	0.002
Education	0.031	0.182	0.868	0.134
Ability to work	0.000	0.000	0.000	0.000
Adjusted R^2^	0.161	0.225	0.171	0.256

## Data Availability

Data on the Finnish basic income experiment is available at the Finnish Social Science Data Archive (https://www.fsd.tuni.fi/en/, accessed on 10 January 2023).

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
