# Peer review of "Basic Income and the Status of Women in an Established Gender-Equal Welfare State: Results from the Finnish Basic Income Experiment"

_ijerph, 2023, doi:10.3390/ijerph20031733_

Round 1

Reviewer 1 Report

The topic selected for the study is interesting..Authors can look in to the Following points. *

* In the introduction segment, Authors can check the flow , where in they start from background information related to the topic and after that they can precisely state the Research problem and significance of the study 

* Lots of relevant  literatures are available related to this topic , Authors can sketch few more relevant literatures 

* The study is based on the data collected at the end of 2018 , As a reader , i am very curious about Effect of Covid 19 and UBI.If authors can bring the data during Covid 19 , The study will be having more practical implications

* Discussion part has to be more precise and Clear 

* Seperate heading or seperate paragraph can be brought in after conclusion for Scope further Research.

* English language Correction is required for the paper 

Kindly carry out following corrections 

Author Response

Reviewer 1:

Thank you for your constructive comments that we have taken into consideration as much as we have been able to do.

  • The manuscript is not proofread by Cambridge Proofreading.

  • In the introduction segment the flow is improved and background information related to the topic is separated. Text is shortened. Research questions are reformulated:

1)         Whether UBI enhances emancipation?

2)         Whether UBI enhances employment?

3)         Are there gender-specific effects?   

Significance:

we write   “UBI has not previously tested in a well-developed welfare state with residence-based basic security and comprehensive social insurance programs. The Finnish experiment was the first one in such a context. There are no previous studies that have analysed Finnish experimental data to decompose possible gender effects.”

  • Authors can sketch few more relevant literatures:

Indeed, there are lots of publications. We have tried to include the ones that are central for our story line. We have added a number of new references as follows:

Calnitsky, D. (2016). ‘“More normal than welfare”: the Mincome experiment, stigma, and community experience’, Canadian Review of Sociology 53(1), 26–71.

Calnitsky, D. ( 2018), “Basic Income and the Pitfalls of Randomization”, Contexts Volume 18 Issue 1.

Cox, E. (2019) “Feminist perspectives on basic income, in E. Klein, J. Mays, T. Dunlop (Eds.), Implementing a Basic Income in Australia: Pathways Forward, Springer International Publishing, Cham (2019), pp. 69-85

Forget, E. Basic Income for Canadians: From the COVID-19 Emergency to Financial Security for All. James Lorimer: Toronto 2020.

Hum, D., and Simpson, W. Economic response to a guaranteed annual income: Experience from Canada and the United States. J of Labor Economics 1993,11, 263–96.

Hum, D.; Simpson, W. A guaranteed annual income: From Mincome to the millennium’, Policy Options 2001, 22 78–82.

McKay, A. Rethinking Work and Income Maintenance Policy: Promoting Gender Equality Through a Citizens’ Basic Income. Feminist Economics 2001, 5, 97-118.

McKay, A. Why Citizens’ Basic Income. A Question of Gender Equality of Gender Bias. Work, Employment and Society, 2007, 21, 337-3348.

Standing, G. Basic Income: And How We Can Make It Happen; Penguin Books: London 2017.

Standing, G. Battling Eight Giants: Basic income now. Tauris: London 2020. 

van Parijs, P.; Vanderborght, Y. Basic Income: A Radical Proposal; Harvard University Press: Cambridge Massachusetts & London 2017.   

  • Effect of Covid 19 and UBI. If authors can bring the data during Covid 19 , The study will be having more practical implications.

The experiment ended before the Covid-19 pandemic: Therefore, unfortunately, we do not have data on the effects of the pandemic.

  • Discussion part has to be more precise and Clear. Seperate heading or seperate paragraph can be brought in after conclusion for Scope further Research.

We have now divided the concluding section as follows: Conclusions, Too universalising discussions and Limitations of the study and future research.

Reviewer 2 Report

This is a very interesting and relevant study, with an original topic.

  1. The abstract should be rewritten so that it sounds more scientific.
  2. The introduction section is well-argued. However, I will suggest splitting this section into ‘Introduction’ and ‘relevant literature’. Also, the authors should re-write the contribution of the study. I recommend the authors see the following paper.
    https://doi.org/10.1007/s11135-020-01046-x
  3. The scientific relevance of the study is not stated (What is the gap in the literature that is addressed by this study?) Authors should update the literature. See the following articles and cite them.
    https://doi.org/10.22024/UniKent/03/fal.51

https://nottingham-repository.worktribe.com/output/1015984

https://doi: 10.1080/23311886.2019.1707005
https://doi.org/10.1016/j.ecolecon.2021.107152

https://doi.org/10.1017/S0047279421000519

  1. Have the authors considered the theoretical framework or any theories when conducting the research? And also suggest formulating the study hypotheses.
  2. The methodology section is sound.
  3. The results are well explained. 
  4. There is no discussion on the novelty that the study would like to add to the literature. It is suggested to add theoretical and practical implications of the study.

Author Response

Reviewer 2.

Thank you for your positive and constructive comments that we have tried to satisfy as much as we could.

The text is corrected by Cambridge Proofreading.

  • The abstract should be rewritten so that it sounds more scientific.

We have rewritten the abstract and added more precise description on the methods and data and omitted some more speculative parts.  We write:

“On the one hand, UBI is seen as a strong incentive for women to stay at home and be perma-nently locked into their care responsibilities. On the other hand, UBI is seen as a tool for em-powerment, to increase women’s autonomy, fortify their capacity to act, and guarantee their individual income and income security. This paper contributes to these debates by asking if UBI enhances women’s empowerment or not. Using the survey data compiled in the context of the Finnish basic income experiment (2017–2018), we compare survey responses from the UBI treatment group (n = 586) and the control group (n = 1,047). Our results based on χ² statistics and regression analyses show that, while UBI did not affect employment, it was positively associated with individual capacities and confidence in various aspects of life. However, these empowering effects were universal and did not differ between women and men. Our results indicate that UBI is not a gender equality-related issue in established gender-equal Nordic welfare states.”

  • The introduction section is well-argued. However, I will suggest splitting this section into ‘Introduction’ and ‘relevant literature’. Also, the authors should re-write the contribution of the study.

 We have rewritten the intro and background section and shortened them. We have separated Introduction and the section called  “Basic income and gender in previous studies” that includes relevant literature on the topics. Furthermore, we have added a substantial number of new references.  

Regarding contribution, we write:  “UBI has not previously tested in a well-developed welfare state with residence-based basic security and comprehensive social insurance programs. The Finnish experiment was the first one in such a context. There are no previous studies that have analysed Finnish experimental data to decompose possible gender effects.” Contribution is also emphasised in the abstract. In the concluding section we criticise previous discussion for being too universalising. Context matters.

  • Have the authors considered the theoretical framework or any theories when conducting the research? And also suggest formulating the study hypotheses.

The aim of the study is to evaluate previous studies and arguments on UBI and to use empirical data from the experiment that up to now is the biggest randomised field experiment with obligatory participation. Thus, our goal is more mundane than evaluate theories. This said, there are implicitly theories and hypotheses included. In a way, various statements and arguments in the UBI debate are our hypotheses. They are formulated in our research questions. We have also made Sen’s and Nussbaum’s capability approach more visible as follows:

“In our case, emancipation is rated as empowerment, that is, people’s capacities for action in their own lives in line with Sen’s and Nussbaum’s theories on capabilities [17, 18, 19]. According to them, people must have the capabilities to master their own lives and participate customarily in the society they live. Nussbaum [19] separates two different forms of capabilities. While, the first one, i.e. the internal capability is linked to the individual's own agency, the second one, ‘combined capability’, is related to societal institutions enabling individuals to use their internal capabilities. The proponents of UBI argue that UBI is a policy measure to fortify both forms of capabilities. Thus, UBI is a device for emancipation. Our main goal is to analyse if that hypothesis holds or not. We ask:

1)                         Whether UBI enhances emancipation?

2)                         Whether UBI enhances employment?

3)                         Are there gender-specific effects? 

 Capabilities are discussed in the empirical part and in the concluding section.  

  • There is no discussion on the novelty that the study would like to add to the literature. It is suggested to add theoretical and practical implications of the study.

Regarding novelty, the mainstream UBI discussion seems to rather blind on the role of social, health care, employment and educational services. We write: “However, this is not a sufficient condition to achieve these goals. In addition, we need an amplitude of family policy measures, such as available and affordable childcare, and we need well-functioning healthcare, education, and employment services to support people and societies in their path towards gender equality. Hence, gender equality cannot be achieved by a single social policy measure. The achievement of more equal societies re-quires several measures that consistently seek to promote that value. A decent level of money is a necessary but insufficient condition for empowering people.”   

We thank the reviewer for these nice comments: The methodology section is sound. The results are well explained.

Reviewer 3 Report

1) At Line 9, the word “oppos-ing” must be written as “opposing”

2) The authors should mention the limitation(s) of the study in the Conclusion section

3) The authors used the British English. If the American English is mandatory, then they must replace the following words in the paper:

- “labour” with “labor”

- “analyse” with “analyze”

- “favour” with “favor”

- “behaviour” with “behavior”

- “centre” with “center”

- “randomised” with “randomized”

Author Response

Reviewer 3.

Thank you for your positive and constructive comments that we have tried to satisfy as much as we could.

The text is corrected by Cambridge Proofreading.

Text is written and proofread in British English.

The concluding section is rewritten and reformulated. There is now a separate section on the limits of the study and some deliberations on future studies.

Round 2

Reviewer 1 Report

Authors have carried out suggestion forwarded by the reviewers and it is visible in paper as well . Congratulations